Comparative genomics of the black rot pathogen Xanthomonas campestris pv. campestris and non-pathogenic co-inhabitant Xanthomonas melonis from Trinidad reveal unique pathogenicity determinants and secretion system profiles

http://orcid.org/0000-0002-0937-0046 Ramnarine Stephen D. B. Jr. stephen.ramnarine@my.uwi.edu
Jayaraman Jayaraj
Ramsubhag Adesh Adesh.Ramsubhag@sta.uwi.edu
Department of Life Sciences, The University of the West Indies , St. Augustine , Trinidad and Tobago
Venancio Thiago
Electronic publication date: 2022 Jan 3
Publication date: 2022
Volume: 10
Electronic Location ID: e12632
Received 2021 Jul 27; Accepted 2021 Nov 22
Copyright: © 2022 Ramnarine et al.
Copyright year: 2022
Copyright holder: Ramnarine et al.
License: This is an open access article distributed under the terms of the Creative Commons Attribution License, which permits unrestricted use, distribution, reproduction and adaptation in any medium and for any purpose provided that it is properly attributed. For attribution, the original author(s), title, publication source (PeerJ) and either DOI or URL of the article must be cited.
License URL: https://creativecommons.org/licenses/by/4.0/

Keywords: Xanthomonas campestris pv. campestris, Xanthomonas melonis, Genome plasticity, Integrated mobile elements, Type 3 secretion system, Bacterial secretion systems

Funding: The University of the West Indies, St. Augustine, Trinidad ACP-EU Research Project #17069 This work was supported by the Campus Research and Publication Fund (CRP.5.APR17.45) of The University of the West Indies, St. Augustine, Trinidad and the ACP-EU Research Project Grant (#17069) awarded to J.J and AR. The funders had no role in study design, data collection and analysis, decision to publish, or preparation of the manuscript.

==============================
Black-rot disease caused by the phytopathogen Xanthomonas campestris pv. campestris (Xcc) continues to have considerable impacts on the productivity of cruciferous crops in Trinidad and Tobago and the wider Caribbean region. While the widespread occurrence of resistance of Xcc against bactericidal agrochemicals can contribute to the high disease burdens, the role of virulence and pathogenicity features of local strains on disease prevalence and severity has not been investigated yet. In the present study, a comparative genomic analysis was performed on 6 pathogenic Xcc and 4 co-isolated non-pathogenic Xanthomonas melonis (Xmel) strains from diseased crucifer plants grown in fields with heavy chemical use in Trinidad. Native isolates were grouped into two known and four newly assigned ribosomal sequence types (rST). Mobile genetic elements were identified which belonged to the IS3, IS5 family, Tn3 transposon, resolvases, and tra T4SS gene clusters. Additionally, exogenous plasmid derived sequences with origins from other bacterial species were characterised. Although several instances of genomic rearrangements were observed, native Xcc and Xmel isolates shared a significant level of structural homology with reference genomes, Xcc ATCC 33913 and Xmel CFBP4644, respectively. Complete T1SS hlyDB, T2SS, T4SS vir and T5SS xadA, yapH and estA gene clusters were identified in both species. Only Xmel strains contained a complete T6SS but no T3SS. Both species contained a complex repertoire of extracellular cell wall degrading enzymes. Native Xcc strains contained 37 T3SS and effector genes but a variable and unique profile of 8 avr, 4 xop and 1 hpa genes. Interestingly, Xmel strains contained several T3SS effectors with low similarity to references including avrXccA1 (~89%), hrpG (~73%), hrpX (~90%) and xopAZ (~87%). Furthermore, only Xmel genomes contained a CRISPR-Cas I-F array, but no lipopolysaccharide wxc gene cluster. Xmel strains were confirmed to be non-pathogenic by pathogenicity assays. The results of this study will be useful to guide future research into virulence mechanisms, agrochemical resistance, pathogenomics and the potential role of the co-isolated non-pathogenic Xanthomonas strains on Xcc infections.

Introduction

Xanthomonas is one of the most important and widespread genera of phytopathogenic bacteria which infects most commercial food crops (cabbage, citrus, tomato, peppers, and rice), trees, and ornamental plants (Ryan et al., 2011; Jacques et al., 2016; An et al., 2020). Being a successful, versatile, and globally distributed pathogen (Lamichhane et al., 2018) armed with a robust genomic arsenal, this genus requires constant attention to develop genomic atlases to identify features that influence complex host-pathogen interactions. Pathoadaptations afford multiple functions and capabilities, including host defence evasion, expression of a diverse repertoire of protein and DNA effector genes (transcription activator-like effectors (TALE’s)), degradative enzyme and biofilm production, and utilization of multiple secretion systems (Ryan et al., 2011; Vicente & Holub, 2013; Alvarez-Martinez et al., 2021). The ability to exchange and maintain mobile elements within the wider Xanthomonad family further increases its genomic fortitude with the ability to encode for copper, heavy metal, and agrochemical resistance factors (Ryan et al., 2011; Behlau et al., 2013). Of particular interest to the current study is native copper resistant Xanthomonas campestris pv. campestris (Xcc), the causal agent of black rot disease of crucifiers, and the co-isolated copper resistant Xanthomonas melonis, which has never previously been associated with crucifers. Originally part of the X. campestris group, Xmel is reported as a soft rot pathogen affecting melons but not much is known beyond this (https://www.cabi.org/isc/datasheet/56949, (Vauterin et al., 1995)). Both species have the potential to affect disease management via agrochemical resistance and potential dual-species plant-microbe interactions.

Of the ~3,736 Xanthomonas genomes available in GenBank, Xcc contributes only a small number (87) and is dominated by others including X. phaseoli (161), X. citri (216), X. arboricola (144), X. oryzae (457), and the tomato pathogens X. vesicatoria, X. perforans and X. euvesicatoria (239). Only 1 Xcc genome from the database originated from the Caribbean and it was from the strain BrA1 isolated from Trinidad (Lugo et al., 2013) while only 1 Xmel assembly exists in the GenBank as of September 2021. While Xanthomonas strains and closely related species display genomic collinearity, rearrangements and a diverse mobilome are common (Comas et al., 2006; Ryan et al., 2011). Mobilome factors, including IS elements, transposons, and plasmids, contribute to horizontal gene transfer (HGT), translocation, and acquisition of patho-capacities including LPS, Type 3 and 4 secretion systems (T3SS and T4SS respectively), and resistance elements (Vicente & Holub, 2013; An et al., 2020).

Virulence and pathogenicity in Xcc are influenced by the secretion systems, effectors, extracellular enzymes, and polysaccharides, which in turn affect the host range and characteristics of disease (Lu et al., 2008; White et al., 2009). Of particular importance to pathogenicity in Xcc are the Types 2 (T2SS), 3, and 5 secretion systems (T5SS) (Vicente & Holub, 2013). Although Type 1 (T1SS), Type 4 (T4SS), and Type 6 (T6SS) (partial) are also present, their strong links to virulence are yet to be established. The T3SS represents the most important pathogenic determinant, and its function includes evading host defence and translocating effectors into host cells (Büttner & Bonas, 2010). The T3SS pathogenicity island falls into the Hrp2 family T3SS (An et al., 2020), comprising of hrp (hypersensitivity response and pathogenicity), hrc (hrp-conserved), and hpa (hrp-associated) genes (Cornelis & Van Gijsegem, 2000; da Silva et al., 2002; Tampakaki et al., 2004). Associated secreted T3 effector diversity is highly correlated with host adaptation and is attributed to ~39 different Xop and Avr proteins (White et al., 2009). The T2SS secretes proteins from the periplasm to the extracellular milieu such as proteases, lipases, toxins, and plant cell wall degrading enzymes (PCWDEs). These enzymes, including endoglucanases, xylanases, cellulases, pectinases and cellulases, are thought to be essential in overcoming host defences, movement across physical barriers to other plant areas, and nutrient acquisition in limited states (Lu et al., 2008; Solé et al., 2015).

In Xcc, the T1, 4, and 5SS do not appear to play a role in virulence as they do in other species (He et al., 2007; Vicente & Holub, 2013; Alvarez-Martinez et al., 2021). However, the T1SS is linked to virulence in X. oryzae pv. oryzae and typically translocate effector proteins including bacteriocins, RTX-nonapeptide motif-containing proteins (possible toxins), and adhesins (Jarrod, Sondermann & O’Toolea, 2018). The T4SS translocates proteins and DNA-protein complexes into foreign cells as an antagonistic or conjugative response (Souza et al., 2011). Currently, this secretion system is split into the T4ASS (typical of Xanthomonas) and the T4BSS (typical of human pathogens) (Guglielmini et al., 2014). In conjunction with the T4ASS chromosomal genes, the T6SS in Xanthomonas appears to be involved in an antagonistic response towards other bacteria and amoeba (Bayer-Santos et al., 2018).

While not directly involved in virulence, the production of biofilm, EPS, and adhesions play a critical role in establishing colonies and adherence to host surfaces within a complex microbial community, which are essential for disease initiation (Cesbron et al., 2015). In Xcc, biofilms are essential for survival in-planta, on leaf surfaces, disease progression, binding of Ca2+ and other divalent ions, and inhibition of callose deposition (Aslam et al., 2008). Additionally, LPS (lipopolysaccharides) O-antigens, an integral membrane component necessary for survival, have been linked to virulence. LPS biosynthesis is controlled by >20 genes (Braun et al., 2005) with the entire biosynthetic cluster highly variable among species (Vorhölter, Niehaus & Pühler, 2001; Patil & Sonti, 2004; An et al., 2020).

The current study addresses the lack of representative Xcc genomes native to Trinidad and adds vital genome characteristics of these pathogenic strains to worldwide databases. From this study, genomes of pathogenic Xcc and non-pathogenic co-inhabitant Xanthomonas melonis (Xmel) have been sequenced and characterized. Although a previous study characterized a single Xcc strain, BrA1 (Behlau et al., 2017), analysis of multiple strains has not been attempted before. While not much is known on Xmel, the current study contributes to the knowledge on the ecological role of this species. The study presents detailed evidence of genome plasticity and virulence gene cluster diversity in native species. The data will be useful for future studies on specific genomic clusters related to pathogenicity, virulence, horizontal gene transfer and recombination events.

Materials and Methods

Bacterial strains used

Morphologically identified and PCR-validated Xcc isolates were selected from a larger collection made from 2015–2017 (unpublished) at the UWI-St. Augustine- Plant-Microbe laboratory. These isolates were obtained from infected crucifer plants from fields that were affected by black-rot disease despite the heavy use of copper-based fungicides. Bacteria were isolated from infected crucifer leaf tissue according to methods described by Lugo et al. (2013), and morphologically identified as Gram-negative, and displaying mucoid, convex, yellow colonies on Nutrient Agar. Copper sensitivity was determined using CuSO4 amended MGY agar (Behlau et al., 2012), according to the protocol outlined by Marin et al. (2019). Nine strains from three major crucifer cultivation areas in Trinidad with representative copper sensitivity profiles were chosen for whole-genome sequencing. The copper resistant native Xcc BrA1 characterised elsewhere (Behlau et al., 2017) was also included and re-sequenced in this study.

Genomic DNA preparation and Illumina sequencing

A modified CTAB protocol (Lugo et al., 2013) was used to extract total genomic DNA. DNA quality assessment, library preparation, and sequencing were carried out at Novogene Corporation (USA). High MW intact DNA (≥10 ng/µL) was assessed for quality and quantity by agarose gel electrophoresis and the use of a Qubit 2.0 fluorometer. The DNA samples were then fragmented using sonication, followed by end polishing, ligation to Illumina adaptors, and amplification with P5 and P7 index oligos. The AMPure XP system was used to purify amplified products and libraries were built using the NEBNext Ultra II DNA Library Prep Kit with a ~350 bp insert size. The size distribution and concentration of libraries were assessed using an Agilent 2100 Bioanalyzer and by qPCR. Illumina sequencing was performed on a Hiseq system (150 bp PE) to a depth of 1G. Adapters and ligation sequences were removed, and raw sequences were filtered to provide reads at Phred scores >30 and error ~0.03%.

Whole genome assembly, annotation, and species identification

Bioinformatic manipulations were carried out on the public server usegalaxy.eu (Afgan et al., 2018) unless otherwise stated. Read pair association and quality were assessed using FastQC 0.72 and trimmed with Cutadapt 3.4 to maintain a QC ≥30 and length ≥100 bp. Trimmed reads were assembled using Shovill 1.1.0 (Spades option with pilon enabled). Resultant multi-fasta contigs were annotated using Prokka 1.14.5 and RAST 2.0 (Overbeek et al., 2014). Annotated 16S rRNA gene sequences were then analyzed using the RDP Classifier (Wang et al., 2007). The Microbial genome atlas (Rodriguez-R et al., 2018) and GTDB-Tk 1.1.0 (Kbase) (Chaumeil et al., 2020) tools were used to determine species identity and to assess genome contamination.

The virulence capabilities of native Xanthomonas strains were assessed to further identify isolates. Cabbage (Tropicana variety) seedlings at the 2-week stage were sown in a 2:1 PRO-MIX and manure/potting mix combination in 16oz pots under open field conditions. Plants were fertilized at recommended levels as outlined by the Ministry of Agriculture, Trinidad and Tobago and irrigated daily. At 4 weeks old, uniform growth seedlings with at least 6 full leaves (past the 2-week stage) were inoculated under humid conditions using the leaf clipping method outlined by Lugo et al. (2013). Xcc and Xmel cultures were grown for 48 h in NB and standardized to an OD600 of 1 before use. Three plants were used per isolate and 3 leaves were clipped per plant on either side of the middle vein. Disease symptoms were assessed at 10 days post infection. Observed results and lesion measurements are contained in Table S11.

Genome completeness and general statistics were obtained using BUSCO 5.0 and QUAST 5.0 (Mikheenko et al., 2018). Assembled genomes for all 9 native isolates were submitted to NCBI under BioProject PRJNA701249. Whole-genome SNP-Phylogenetic reconstruction with Xanthomonas reference genomes was carried out using CSIPhylogeny (Kaas et al., 2014) and FastTree 2.0 (GTR, SPR, 1000 bootstrap) (Price, Dehal & Arkin, 2010) via NGPhylogeny (Lemoine et al., 2019). Phylogenetic trees were edited in TreeGraph 2 (Stöver & Müller, 2010). Accurate sequence typing and grouping were further carried out using rMLST typing via PubMLST Species ID (Jolley et al., 2012).

Genome characterization and feature annotation

Genomic features of each strain were obtained from the RAST server. Pangenome analysis for core genes (95% similarity) and accessory genes (<50% strains) were determined using the Roary 3.13.0 pipeline (Page et al., 2015). Pan-genome functional categorization using COG (Cluster of orthologous groups) classification was described using EggNog mapper 2.0 (Huerta-Cepas et al., 2017) and the NCBI COG database (https://www.ncbi.nlm.nih.gov/research/cog/). Identification of potential secondary metabolite biosynthetic gene clusters (BGC’s) was done using the AntiSMASH 5.0 webserver (Blin et al., 2019). Genome alignments to determine regions of similarity, gain or loss of segments, and recombination events were carried out using progressiveMauve (Darling, Mau & Perna, 2010). Draft genomes were first ordered against the reference strains, X. campestris pv. campestris ATCC 33913 and X. melonis CFBP4644, using the Mauve Contig Mover (MCM). Secretion systems were identified based on cumulative annotations from Prokka and RAST, predictions using the EffectiveDB server (Eichinger et al., 2016), tblastn and blastn (>70% coverage and identity) analysis using manually curated databases of published sequences.

Genes for plant cell wall degrading enzymes (PCWDEs) were identified using the dbCAN2 web server (Zhang et al., 2018). Identification of mobile genetic elements was facilitated from annotated CDS and the ISFinder database (Siguier et al., 2006). The prediction of potential integrated and conjugated mobile elements was facilitated using ICEberg 2.0 (Liu et al., 2019). The location of mobile elements in each native strain was generated using blastn against ordered, concatenated contigs. Circular plots were then generated using this data and the CGView server (Grant & Stothard, 2008). Plasmid derived contigs were predicted using PlasmidVerify (Antipov et al., 2019), a Bayesian classifier trained on HMM’s associated plasmid sequences, and RFPlasmid (van der Graaf van Bloois, Wagenaar & Zomer, 2020), an implementation of a Random Forest model on multiple bacterial DNA species databases. Predicted plasmid derived contigs >1 kb were identified using both PLSDB (Galata et al., 2019), and blastn (%ID >80%) against a manually curated plasmid database (https://ftp.ncbi.nih.gov/refseq/release/plasmid/). The CRISPRCasTyper 1.4.1 server (Russel et al., 2020) was used to identify CRISPR arrays and Cas proteins, with identified spacers subjected to Blast analysis against the NCBI phage, plasmid, and IMGVR databases via the CRISPRTarget tool (Biswas et al., 2013). Heatmaps were generated using TBtools 1.082 (Chen et al., 2020) and synteny maps using Gene Graphics (Harrison, Crécy-Lagard & Zallot, 2018).

Results

Genome statistics of native Xanthomonas isolates from Trinidad

Nine Xanthomonas contig assemblies were generated from black-rot leaf lesion isolates with varying levels of copper tolerance/resistance using the Illumina Hiseq shotgun sequencing approach. To validate the identity in our collection, Xcc strain BrA1 (provided by Prof. Jeffery Jones, University of Florida, USA) was re-sequenced. General assembly metrics are presented in Table S1. Overall, 8.6–21 M reads with a Phred score of ~36 were obtained after quality filtering, yielding 107–257 de-novo assembled contigs (44–90 >1 kb). Two Xanthomonas species were identified with 98% ANI similarity to reference genomes using MIGA and GTDB-Tk (Table S1). Xcc strains included Ar1BCA1, Ar1PC2, BrA1, CaNP1C, Cf3C and Cf4B1, while CaNP1D, CaNP5B, CaNP6A and DMCX were classified as Xmel. All assembled genomes were 99.8% complete (BUSCO), showed typical GC% (65.1 to 66.4%) for each species, and were supported by 251–615X coverage (Table 1). The assemblies also had complete single 16S rRNA genes and less than 0.01% of reads were identified as contaminants, which were removed before downstream analysis. Pathogenicity against the tropicana cabbage variety was confirmed for all Xcc strains but Xmel isolates did not elicit any symptoms (Table S11).

Table 1 De-novo shotgun assembled genome statistics of Trinidadian strains of two Xanthomonas species isolated from infected crucifer leaves.

Strain ∆	Copper sensitivity†	Sequenced length (kb)	GC%	Coverage (X)	N50	rMLST ST‡	NCBI Accession	Total genes	Coding genes	RNA genes	tRNAs	ncRNAs	Total pseudogenes	
Species: Xanthomonas campestris pv. campestris (Xcc)	
Ar1PC21	Tolerant	5.23	65.01	367	166,424	rST-171941	JAFFQP000000000	4,621	4,310	150	53	94	161	
Ar1BCA11	Tolerant	4.98	65.2	274	130,896	rST-132382	JAFFQL000000000	4,374	4,100	139	53	84	135	
BrA12	Resistant	5.15	65.08	615	130,897	rST-132382	–	4,410	4,283	153	59	94	138	
Cf3C3	Resistant	5.16	65.06	296	113,525	rST-132382	JAFFQN000000000	4,575	4,292	140	53	84	143	
Cf4B13	Resistant	5.16	65.07	317	129,122	rST-132382	JAFFQM000000000	4,563	4,278	140	53	84	145	
CaNP1C3	Tolerant	5.08	65.03	251	176,946	rST-68128	JAFFQO000000000	4,510	4,228	144	52	86	138	
Species: Xanthomonas melonis (Xmel)	
CaNP1D3	Tolerant	4.78	66.04	579	166,537	rST- 171942	JAFFQH000000000	4,120	3,938	99	52	44	83	
CaNP5B3	Resistant	4.84	65.99	276	156,120	rST- 171943	JAFFQJ000000000	4,182	3,980	102	56	43	100	
CaNP6A3	Resistant	4.84	65.99	335	172,216	rST- 171942	JAFFQI000000000	4,202	3,998	105	56	43	99	
DMCX4	Tolerant *	4.8	65.97	386	282,115	rST- 171942	JAFFQK000000000	4,195	3,990	103	56	44	102	
Notes:

∆ - superscripts represent source crops: 1; Bok Choy, 2; Broccoli, 3; Cauliflower, 4; Cabbage.

† Copper Tolerant isolates grew in the presence of up to 200 ppm CuSO4, while resistant strains grew in the presence of up to 360 ppm CuSO4.

‡ rST of Ar1PC1 and all Xmel represent unique sequence types newly deposited in the PubMLST database, full allelic profiles are given in Table S2.

* Xmel strain DMCX was only screened up to 200 ppm CuSO4.

Phylogenetic relationships among native isolates and Xanthomonas reference species were assessed using common whole-genome SNPs as seen in the cladogram in Fig. 1. Native Xcc isolates clustered with the type strain (ATCC 33913) and other RefSeq genomes (X. campestris cluster) while showing separation according to rMLST within this clade. A similar placement is seen with Xmel strains, where all native strains were grouped with the single reference assembly (CFBP4644). The rMLST typing of native isolates based on 53 ribosomal rps and rpl genes (Table S2) showed that the Xcc strains fell into three groups based on ribosomal allelic profiles (rST) (Table 1). Four of the 6 isolates (Ar1BCA1, BrA1, Cf3C, Cf4B1) together with one reference (Xcc 3811) were categorized as rST-132382, while one isolate (CaNP1C) and five references (including MAFF302021) had the rST-68128 profile. The remaining isolate Ar1PC2 did not match with any existing rST and has been added to a new group on the database. All native Xmel strains were classed as unique, and the newly identified rST were added to the database. Two rST’s were inferred from the four native Xmel isolates, where CaNP1D, CaNP6A and DMCX fell in the same group.

Figure 1 Phylogenetic reconstruction of whole genome SNPs of native Xanthomonas strains and reference genomes (cladogram).

Native strains are represented in purple text with their respective rMLST in brackets. Called SNPs from 46 reference species and 10 native genomes were concatenated by CSIPhylogeny and ML Phylogenetic reconstruction carried out using FastTree with 1000 boostrap (SH-Like) replicates. Reference genomes consisted of RefSeq representative and/or next available assembly genomes from the full NCBI taxonomic listing of relevant Xanthomonas spp.

Progressive Mauve genome alignment, pangenome analysis, and Biosynthetic gene cluster (BGC) prediction

The annotated genome statistics (Table 1) show that not only were the Xcc genomes larger, but apart from tRNAs, they also contained more CDS than native Xmel isolates. To maximize local collinear block (LCB) size and arrangement, multi-fasta assemblies were ordered against the most curated reference genomes. Collinear LCB’s among strains within species, as well as distinct patterns of rearrangements via inversions and translocations within contigs, were identified (Figs. 2 and 3).

Figure 2 Progressive Mauve alignment of Xcc genomes of native isolates ordered against the reference Xcc ATCC 33913.

Figure 3 Progressive Mauve alignment of Xmel genomes of native isolates ordered against the reference Xmel CFBP4644.

The Xcc and Xmel genomes differed to such an extent that combined alignments for both species were not possible. The genomes of the native Xcc isolates had a higher level of conserved structural organization and fewer rearrangements compared to Xmel. This was supported by the congruent arrangement and mostly complete similarity plots of each Xcc LCB. Furthermore, these differences in similarity plots and gaps between LCB’s (incomplete/white regions) indicate that there are more cases of unique genomic regions within native strains than the reference. Xmel genomes had a larger number of smaller LCBs with a more complex arrangement.

The pangenome of each species contained 5,019 and 4,491 genes respectively. Approximately 83% of genes made up the core genome for each species (90–100% occurrence in strains within species), with ~900 accessory genes having <50% occurrence across strains. CDS were found in ~94% and ~62% of core and accessory genes, respectively, with 10% and 27% remaining unclassified. Figure 4 depicts COG categories that provide functional descriptions of the core genome, with major functions including energy production, amino acid, lipid, and cell wall biosynthesis and metabolism, inorganic ion transport and metabolism and signal transduction. Except for secondary metabolite biosynthesis and metabolism, as well as intracellular vesicular transport, the Xcc genomes had more genes in all categories. High copy numbers of genes in both species were functionally classified as outer membrane Fe transporters, MDR/outer-membrane efflux and transporters (TolC, AcrA), transcriptional regulators (LysR, AcrR, OmpR, LacI/PurR, NarL/FixJ), NAD(P)-dependent dehydrogenases, MFS family permeases, signal transduction histidine kinases, chemotaxis proteins (MCP, CheY), GGDEF domain secondary messengers, cell wall biosynthesis, SAM-dependent methyltransferases, glycosyltransferases, and detoxification enzymes (See Table S3 for full details).

Figure 4 COG functional characterization of native Xanthomonas pangenome elements.

Accessory genes of both species were more functionally variable than core genes, with Xcc strains dominating in most categories. Mobile elements made up a large portion of the pangenome but were more prevalent in the accessory genome. COG classification of accessory genes revealed that they were involved in extracellular stress response, transcription, DNA repair and recombination, and cell motility. In Xcc strains, a greater number of accessory genes were involved in intracellular trafficking and secretion, and secondary metabolite transport. Other Xcc gene-specific functions included Par based chromosome segregation and plasmid partitioning, NRPS enzymes, Fe transport receptors, and T4SS proteins involved in mobile pilus biogenesis. In Xmel, the accessory genes were mostly involved in putative lipase activity, biofilm formation, adhesion, Fe transport, and type II secretion system (T4SS, T2/4SS ATPases, GspE) (Table S3). Most functional categories and members in the accessory genome were also found in the core genome indicating possible functional redundancy.

Table 2 summarizes the BGC and predicted products based on syntenic similarity to reference Xanthomonas genome clusters in the MiBIG database as output from the AntiSMASH webserver. The number of BGC’s in Xmel genomes (4–14) was higher than in Xcc genomes (3–8). All Xanthomonas genomes contained the hallmark BGC’s relating to xanthoferrin and xanthomonadin production. Differences in other BGC’s include the production of pseudopyronine seen only in Xcc BrA1, while production of rhizomide A, B, and C was predicted in three Xmel isolates and Xcc Ar1PC2. Multiple NRPS clusters were found in all strains. While some were homologous to RefSeq NRPS genes from the same or other Xanthomonas species, others appear to be linked to uncharacterised compounds. Interestingly, putative class II lasso peptide clusters were predicted only in Xmel genomes.

Table 2 Predicted Biosynthetic gene clusters (BGC’s) and secondary metabolites in Xcc and Xmel genomes.

Strain	# BGC	BGC type	Predicted metabolites*	
Species: Xcc	
Cf4B1	3	siderophore, NRPS, aryl polyene	Xanthoferrin, Xanthomonadin I, putative NRPS	
Cf3C	3	siderophore, NRPS, aryl polyene	Xanthoferrin, Xanthomonadin I, putative NRPS	
CaNP1C	3	siderophore, NRPS, aryl polyene	Xanthoferrin, Xanthomonadin I, putative NRPS	
Ar1BCA1	3	siderophore, NRPS, aryl polyene	Xanthoferrin, Xanthomonadin I, putative NRPS	
Ar1PC2	8	siderophore, NRPS, aryl polyene	Xanthoferrin, Xanthomonadin I, putative NRPS, Kedarcidin (4%), cyanopeptin (50%), rhizomide A/rhizomide B/rhizomide C (multiple)	
BrA1	3	siderophore, NRPS, aryl polyene	Xanthoferrin, pseudopyronine A/pseudopyronine B (synteny to Xanthomonadin clusters of reference Xcc), putative NRPS	
Species: Xmel	
DMCX	4	siderophore, lassopeptide, NRPS, aryl polyene	Xanthoferrin, Xanthomonadin I, putative lassopeptide	
CaNP6A	12	siderophore, lassopeptide, aryl polyene, NRPS, T1PKS (NRPS)	Xanthoferrin, Xanthomonadin I, putative lassopeptide, xenoamicin A/xenoamicin B (multiple, 25%), xenotetrapeptide, cichopeptin (50%), rhizomide A/rhizomide B/rhizomide C (multiple)	
CaNP5B	14	siderophore, lassopeptide, aryl polyene, NRPS, T1PKS (NRPS)	Xanthoferrin, Xanthomonadin I, putative lassopeptide, xenoamicin A/xenoamicin B (multiple, 25%), xenotetrapeptide, rhizomide A/rhizomide B/rhizomide C (multiple), lokisin (14%), bananamide 1/bananamide 2/bananamide 3 (37%)	
CaNP1D	11	siderophore, lassopeptide, aryl polyene, NRPS, T1PKS (NRPS)	Xanthoferrin, Xanthomonadin I, putative lassopeptide, xenoamicin A/xenoamicin B (25%), xenotetrapeptide, rhizomide A/rhizomide B/rhizomide C, teixobactin (20%), tolaasin A (66%)	
Note:

* Predicted metabolites are based on homology to characterized BGC in the MIBiG database with linked chemical isolation data. Percentages indicate the % similarity to those gene clusters in the database and were 90–100% similar to reference clusters unless otherwise indicated.

Mobile elements and CRISPR-Cas systems

Mobile elements in bacterial genomes represent units of genetic diversity and potentially exogenous gene sequences. Figure 5 shows the distribution of mobile elements detected from RAST annotations and database queries including insertion sequences (IS), transposases and Tn related proteins, plasmid and phage related proteins, recombinases, integrases, and plasmid-derived contigs. The Xcc isolates had overall more genetic elements and more on plasmid-derived contigs than Xmel genomes. Integrases (including phage-derived proteins), a plasmid stabilization protein, and site-specific tyrosine recombinases (XerC, D) were common in both species and most prevalent on chromosomal contigs. The only Xmel species-specific elements were a putative tniA transposase and an un-identified element with a transposase and integrase Pfam domain. Other transposon elements, Tn4652 and Tn4651 tnpT were more prevalent in Xcc plasmid contigs compared to Xmel.

Figure 5 Distribution of annotated mobile elements (non-IS) in Xcc and Xmel chromosomes and plasmid derived contigs (A), and total element numbers in strains from both species (B).

(A) Vertical colour legends refer to gene counts for each respective mobile element occurring on either chromosomal or predicted/mapped plasmid contigs (coloured bar to the left of (A)). (B) Cumulative gene counts for each mobile element in (A), per species.

Identification of plasmid-derived contigs was accepted if contig length was >2 kb and contained >2 plasmid-associated Pfam HMMs with >80% identity to reference plasmid sequences. Sequence % ID and length (kb) of partial or whole contigs mapped to RefSeq plasmid sequences are provided in Tables S4A and S4B. While contigs mapped only to plasmids characterised in Xanthomonas spp., only two homologous plasmids could be reliably predicted in native strains of both species. Native Xcc (BrA1, Cf3C, Cf4B1) contained fragments of NZ_CP018463.1 X. euvesicatoria pLMG930.2 (77–82 kb, 96% ID) whereas Xmel (CaNP5B and CaNP6A) contained 41 kb from NZ_CP018472.1 X. perforans pLH3.1 (99.9% ID).

Seventy-four IS elements (12-plasmid borne, 62-chromosome borne) from eight families (IS407, IS5, IS1031, IS51, IS10, IS3, IS427, IS1595) were found at least once in the genome of native Xanthomonas isolates (Table S5). The IS element families IS1031, IS407, IS5, and IS51 were found in greater abundance in all strains (Fig. 6A). Only chromosomal-based elements were commonly found in isolates of both species, but these same elements were also found on plasmid-derived contigs in Xcc (Fig. 6A). Overall, IS elements outnumbered other mobile elements, both of which were greater in Xcc than Xmel (Fig. 6B).

Figure 6 IS element families and groups present in Xcc and Xmel chromosome and plasmid derived contigs (A), and total IS and other mobile elements per strain (B).

(A) Vertical colour legends refer to gene counts for each respective mobile element occurring on either chromosomal or predicted/mapped plasmid contigs (coloured bar to left of (A)). (B) Cumulative IS elements vs other mobile elements in each native strain from both species.

A greater occurrence of the IS1031 and IS407 family elements was seen in Xcc strains, while IS5 and IS51 family elements were found in greater abundance in Xmel strains. These elements in Xcc strains were homologous to those characterized from X. campestris, X. axonopodis, X. oryzae, and X. fuscans (Table S5). Xmel associated chromosomal IS elements were homologous to those from Azospirillum sp., Azotobacter sp., Burkholderia sp., Janthinobacterium sp., P. aeruginosa; alcaligenes, stutzeri, syringae, Ralstonia solanacearum, S. maltophilia and, X. citri. Figure 7 depicts the location of all mobile elements in each Xcc (A) and Xmel (B) genome. The presence of transposon, IS elements, and integrases in clusters in Xcc genomes suggests a more complex organisation of integrative mobile elements than in Xmel isolates. The latter species’ genome, on the other hand, revealed more instances of IS-rich loci in more than one location. Integrative mobile and conjugative elements were predicted in isolates of both species (using IceBerg2), except strain DMCX, and displayed a lower GC% than other genomic regions (Table S5).

Figure 7 Location and organization of mobile element loci in native Xcc (A) and Xmel (B) genomes.

Crispr-Cas arrays of the Type I-F subtype were only found in Xmel strains (Table 3). The cas1, cas2-3, and csy clusters (csy1-4) were detected preceding a variable length CRISPR region. Direct repeats (DR) of these arrays were 96% identical across the entire region and strains. CaNP5B contained a separate DR entirely that was also part of an orphan array in DMCX. The number of spacers and diversity varied among strains, but the majority overwhelmingly targeted plasmids from X. oryzae, X. hortorum, X. perforans, X. campestris, X. citri; Xylella and Xanthomonas phages, and IMG/VR uncultured phylloplane viruses (IMG/VR).

Table 3 DR sequences of the CRISPR-Cas 1F arrays in native Xmel strains.

Strain	DR sequence	# Repeats	
CaNP1D	GTTCACTGCCGCGTAGGCAGCTCAGAAA	71	
CaNP5B	TTTCTGAGCTGCCTACGCGGCAGTGAAC	124	
CaNP6A	GTTCACTGCCGCGTAGGCAGCTCAGAAA	128	
DMCX	GTTCACTGCCGCGTAGGCAGCTCAGAAA and orphan: TTTCTGAGCTGCCTACGCGGCAGTGAAC	69 and orphan: 62	

T3SS and effector profile of native Xcc and Xmel strains

Native Xcc and Xmel genomes along with complete Xcc references (Table S7) were screened for the presence of 51 T3SS and effector genes (xanthomonas.org; White et al., 2009). Only Xcc isolates had the T3SS and effector genes (Table S8) including avrBS2, PhpE/xopAL1, XccA1/Xca, avrXccA2; hpa4/xopF1, A, B, G/xopA, hpaP; hrcC, J, N, Q, R, S, T, U, hrcV; hrpB1, B2, B5, D, E, F, G, W, hrpX; and xopAM, AY, AZ, D, K, L/LR, N, Q, xopZ1. All native Xcc had one xopX gene with 94–100% similary to Xcc 8004, but Ar1PC2 and references B100, 3811, CFBP 1869, and CN03, also contained another with 62% similarity to Xcc 8004 downstream.

Figure 8 depicts the collinear organization of the conserved ~23 kb hrp T3SS gene cluster of native Xcc strains. The genes in three strains (Ar1BCA1, Ar1PC2, and Cf4B1) were organized in one direction as shown in Fig. 8, whereas those in the remaining isolates (BrA1, CaNP1C, and Cf3C) were oriented in the opposite direction. Interestingly, in all native strains, a Tn element was found downstream of hpa2. Aside from these T3 genes, Fig. 9 illustrates the varied presence and copy number of some avr, xop and, hpaF genes. When using relaxed protein queries, only 4 homologs of Xcc T3 secretion, one avr, and one effector were found in Xmel genomes (Table S9). These genes were <90% similar to Xcc 8004 homologues avrXccA1 (~89%), hrpG (~73%), hrpX (~90%) and xopAZ (~87%). No other T3 secretion and effector proteins were identified in Xmel isolates using homology-based or Pfam hmm-based searches against reference genes. EffectiveDB predictions revealed the presence of potential T3 effectors but only weak support for T3SS function in Xmel genomes. The T3SS from Xcc were predicted to be functional and have a greater number of potential effectors, as shown in Table S10.

Figure 8 hrp gene cluster map of native Xcc strains.

Figure 9 Differential occurrence of selected T3 effectors in native Xcc isolates and GenBank reference genomes.

The different colours in the vertical legend indicate copy number of homologs.

Multiple alignments of Avr, HrpG, HrpX and some Xop proteins were analysed to determine potential variation in effectors shown in Fig. 9 with documented links to virulence (Table 4). Unless otherwise specified, protein sequences were compared to those from reference genome Xcc 8004.

Table 4 Amino acid variation of select T3SS and effector proteins present across native Xcc and Xmel strains.

Protein	Comment	Function	References	
AvrAC/XopAC	All similar (2 substitutions), T358A (except ATCC 33913) and, R70S (ATCC 33913 only)	Mutants infect Chinese cabbage and resistant Arabidopsis Col-0 ecotype	(Xu et al., 2008; Oh, Lee & Heu, 2011)	
AvrBS1	Xcc Ar1BCA1 truncated (321 vs 445). Reference and, Ar1PC2 and CaNP1C share 1 substitution	AvrBs2 -full virulence variable copy number. AvrBS3 - TALE. Korean Xcc cabbage strains (AvrBS1 and 2), Chinese cabbage strains (AvrBs1 and 3), radish strains (AvrBs2).	
AvrBS2	All strains except Xcc ATCC 33913 show an R712K substitution	
AvrBS3	Homology in ~ last 500 aa only and to CN03. Xcc strains Cf3C, Cf4B1, BrA1 similar. Both from Ar1PC2 are different from other native proteins.	
AvrXccB/XopJ5	All similar (2 substitutions- G96V and V197A)	Yop-J like targets host plasma membrane immune suppression (Arabidopsis) not required for full virulence	(Liu et al., 2017)	
AvrXccC/AvrXccAH/xopAH	Native strains contained an additional (identical) 109 aa. Identical except for one substitution (CN03 and Ar1PC2, S12G)	inhibits host immune flg22-callose deposition promotes growth of Xcc in Arabidopsis.	
AvrXccE1	All proteins identical	HR response in mustard cultivars affected by Chinese strains. Putative transglutaminase	(He et al., 2007; Vicente & Holub, 2013)	
AvrXccE2/xopE3	All proteins identical	HR response in Chinese cabbage. Putative transglutaminase	
HrpG	native Xcc identical, native Xmel proteins shorter and not homologous to Xcc	Omp-R type regulator of hrp gene expression.	(Büttner & Bonas, 2010; Feng et al., 2009)	
HrpX	Native Xcc and Xmel proteins longer than references. Xcc homologous but Xmel proteins show multiple substitutions.	AraC-type transcriptional regulator, upregulated by hrpG.	
XopP	proteins from native strains and Xcc 17 appear truncated at the N terminal. Otherwise, identical except for the following substitutions: C85R, D146A, S442R, M566V, I730V	Target ubiquitin ligase in rice and inhibits host enzyme function	(Adlung et al., 2016)	

The T1-6SS, extracellular enzymes, and LPS biosynthesis cluster

The prevalence of T1-6SS and extracellular enzymes found in native Xcc and Xmel genomes are depicted in Fig. 10. Only the T1SS hlyDB gene cluster was found in native genomes. The highly conserved xps and xcs T2SS gene clusters composed of genes C, D, E, F, G, H, I, J, K, L, M, and N were found on separate contigs in all strains of Xcc and Xmel. T4ASS-virB1-11, virD4 genes were also found associated with isolates, but Xcc Ar1PC2 contained the extra virD2 with all but 2 strains containing the virJ gene. Interestingly, 3 Xcc strains possessed T4BSS icm-J/T like genes. Furthermore, all genomes contained the associated and complete T4 pilus gene cluster (pilA-Z) except for Xcc Cf3C which lacked pilI. The pilX gene was also found only in the Xcc isolates. Three T5SS-related genes, xadA, yapH and estA were also detected in both species but only 3 Xmel isolates contained the fhaBC gene cluster. Additionally, while all strains had the T6SS gene tssA, the entire gene cluster tssB-M, and associated hcp, ppkA, and pppA genes were only found in Xmel genomes. The presence of a functional T6SS is supported by EffectiveDB predictions, but only weakly in Xcc strains (data not shown).

Figure 10 Presence of T1-6SS genetic elements and extracellular enzymes in Trinidadian Xcc and Xmel isolates’ genomes.

Blue indicates presence while red indicates absence, “H_” denotes a hypothetical protein.

Native isolates of Xcc (97) Xmel (80) contained putative extracellular carbon targeting enzymes, mainly in the Glycosyl hydrolase (GH) CAZy module with others classified as polysaccharide lyases (PL’s), carbohydrate esterases (CE’s), and those with carbohydrate-binding motifs (CBM’s). PCWDEs targeting plant cell wall components with known SignalP sequences were selected from these results and are depicted in Fig. 10. The enzymes xylanases, glucanases, pectinases, protease, esterases, lipases, and rhamnogalturonases were found in both species, with a higher prevalence in Xcc strains. The Xmel strains lacked the extracellular cellulase repertoire found in Xcc.

The schematic organization of the LPS biosynthesis gene cluster in Xcc B100 with highly conserved flanking genes is shown in Fig. 11A. The LPS cluster is made up of 15 genes that were organized into 3 regions. All Xcc strains had a full gene cluster (Fig. 11B) of ~17.6 kb on a single contig, except BrA1. The WxcB protein sequences in Xcc strains Ar1BCA1, BrA1, Cf3C, and Cf4B1 were up to 60% identical to that of Xcc B100. In Xmel strains, an incomplete LPS biosynthesis cluster was discovered (Fig. 11C), containing 4/15 genes from region 1. In addition, IS401 transposase and other mobile elements were found in this region of DMCX. The LPS biosynthesis clusters in both species’ genomes were flanked by the highly conserved xanAB, rmlDCAB, and metB genes which shared collinearity with Xcc B100.

Figure 11 Schematic representation of the LPS biosynthesis gene cluster present in Xcc (B) and Xmel (C) genomes ordered against the Xcc B100 reference sequence (A).

Only the wxc gene cluster from reference Xcc B100 are showed in (A), flanking genes displayed in (B) are present in this strain.

Discussion

Genomic features of native Xcc and Xmel isolates display a high degree of structural conservation

All isolates were originally obtained from black rot infected crucifer leaf tissue and putatively identified as Xanthomonas spp. This study aimed to characterise pathogenicity and virulence genes, and other genetic factors that allow for survival and adaptation to host and associated environmental ecological niches of native Xanthomonas isolates. Multiple analysis tools were used to confirm the species identity of assembled genomes of native Xanthomonas isolates. This resulted in the identification of 5 Xanthomonas campestris pv. campestris (Xcc) and 4 Xanthomonas melonis (Xmel) isolates. Co-infection of Xcc with other species is not known to occur for black rot disease, and except for cases of secondary infection with soft rot pathogens, the phenomenon is not widely reported in the literature (Vicente & Holub, 2013). While Xcc is characterised as the causal agent of black rot in crucifers (Staub & Williams, 1972) other Xanthomonas campestris pathovars have also been associated with these crops (Vicente et al., 2001; Vicente, Everett & Roberts, 2006). However, there are no reports of Xmel causing disease or being associated with crucifers. Co-infection with different species of Xanthomonas has been described in other host systems where variations in virulence of pathoadapted strains were observed (Cesbron et al., 2015; Bansal, Kumar & Patil, 2020). However, the overall impact of non-pathogenic Xanthomonas as reservoirs of genetic diversity and resources for other species is not well understood but is an important factor to consider in plant-pathogen interactions (Lee, Yang & Huang, 2020; Fernandes et al., 2021). In fact, non-pathogenic species are commonly isolated from many plant sources, but these are overwhelmingly identified as X. aboricola, although at least one study has found non-pathogenic Xcc on crucifer planting material (Lee, Yang & Huang, 2020). This interesting association of Xcc and Xmel in infected crucifer tissues highlights the need to study both pathogenic and potentially non-pathogenic/opportunistic pathogens isolated from the same environment. Analysis of Xmel will help in understanding the ecological role of this species, explore its function as co-inhabitants in the pathogenesis of Xcc, and further add to baseline data on native Xanthomonas isolates.

Multiple metrics strongly supported the identification of native Xanthomonas strains, including ANI and full 16S rRNA gene % ID greater than the species identification threshold (>98%) (Konstantinidis & Tiedje, 2005; Edgar, 2018), and whole-genome SNP phylogenetic reconstruction with reference sequences and rMLST typing (Jolley et al., 2012). The genomes of the Xcc and Xmel isolates had sizes and GC % that were typical for each species. Native Xcc isolates formed three distinct genotypes, with one new rST. The Xmel isolates were grouped into 3 distinct genotypes which only included native isolates, all of which were newly recorded rST’s. Despite belonging to different species with unique genotypes, a core genome with shared COG functions, characteristic of the genus was observed in the Xcc and Xmel isolates. Accessory genes were linked to processes involved in DNA repair and to mobile elements. Such factors point to exogenous DNA and HGT events in native strains which can account for the intraspecific structural and genome size differences. Overall structural genome collinearity was maintained between reference and native Xcc strains, though rearrangements and unique organisation of genomic loci within native strains were observed. Genomic rearrangements are common within Xcc and have been linked to host adaptation within geographic regions (Jacques et al., 2016; An et al., 2020), highlighting the enormous potential of the genus to exist and thrive in different ecological niches. Xmel genomes contained collinear segments and a more chaotic series of arrangements further compounded by the lack of a complete reference genome for this species. However, the small size of LCB’s in both native and the reference assembly, points to greater levels of variation in Xmel compared to Xcc.

Native Xcc and Xmel mobilomes

In both native species, eight IS families were discovered, with Xcc IS elements outnumbering those in Xmel. The IS family complement in Xanthomonas is highly variable displaying species-specific family repertoires with a noticeable reduction in non-pathogenic strains (Parkhill et al., 2001; Cesbron et al., 2015). IS3 and IS5 family elements are commonly found in abundance in Xcc (Thieme et al., 2005; Cesbron et al., 2015), which were also most abundant in native Xcc and Xmel strains. The specific elements in Xcc were first identified in X. campestris, X. oryzae, and other Xanthomonas spp. While those in Xmel were identified from other diverse bacterial genera including Xanthomonas. This repertoire of elements indicates greater instances of HGT between more diverse bacterial species and Xmel strains, where that in Xcc appears to be constrained to closely related species. Mobile genetic elements such as transposable elements (TE’s), phage integrases and extrachromosomal sequences (plasmids) play a substantial role in genome evolution (Dziewit et al., 2012). These elements facilitate autonomous genome restructuring through inversions and enhanced homologous recombination, even with exogenous plasmid sequences (Cesbron et al., 2015). In Xanthomonas, TE’s are linked to the acquisition and modification of resistance and virulence genes (Mahillion & Chandler, 1998; Monteiro-vitorello et al., 2005; An et al., 2020). Of note is the occurrence of IS and other mobile elements flanking the T3SS in Xcc, a phenomenon that adds to the genomic diversity potential of these elements in this genus (Cesbron et al., 2015; Merda et al., 2017; Bansal, Kumar & Patil, 2020). There is a clear capacity for genomic rearrangements and plasticity in both native species. The repertoire and copy number of IS elements have been linked to a more streamlined host-adapted lifestyle in many other bacterial species (Siguier, Gourbeyre & Chandler, 2014). Thus, the greater diversity of these features implies a less host-adapted lifestyle of Xmel.

Tn3 elements, which are commonly associated with the acquisition of antibiotic and heavy metal resistance genes (Nicolas et al., 2015), were most abundant in Xcc genomes while Xmel appears to have unique Tn elements. Integrative conjugative and mobile regions were present in all native strains and displayed characteristic structure, organisation, and lower GC%. These regions in Xcc contained heavy metal transporters, but mostly hypothetical proteins with unknown genomic importance. Exogenous sequences of plasmid origin were predicted with strong support (~98% ID to references) in three (each) Xcc and Xmel genomes from X. euvesicatoria and X. perforans respectively. High homology of these regions to these reference plasmids gives evidence of HGT events. This in combination with the characterised mobilome highlights the complex nature of lateral gene transfer in Xanthomonas. Bacterial genomic phage content and plasmid presence can be influenced by CRISPR-Cas systems. In Xanthomonas, two subtypes (1-C and 1-F) have been found which are also present in other members of Xanthomonadaceae but not in Xcc. Some species also contain complex arrays which tend to target plasmids from other species and phages (Martins et al., 2019). In native strains, only the 1-F subtype CRISPR-Cas system was found in Xmel, with complex arrays targeting diverse plasmid sequences from other species. Its absence has been linked to the presence of multiple diverse plasmids in other Xanthomonas species (Martins et al., 2019) and possibly accounts for the greater number of predicted plasmid contigs in Xcc as well as, the reduced phage content in Xmel isolates.

Pathogenicity island conservation and variable T3 effector repertoire in Xcc

The T3SS pathogenicity island was only detected in native Xcc and was syntenic to reference Xcc islands of average length (~23 kb), with no internal rearrangements. The Xcc effector repertoire comprised 14 common and 12 genes of variable prevalence. These consisted of the 9 core effectors established for this genus (Ryan et al., 2011). Notably, only native strains contained a xopR gene and 3 were missing the xopP gene, both of which are involved in the suppression of PTI (Akimoto-Tomiyama et al., 2012; Üstün & Börnke, 2014). Furthermore, the additional 8 effectors found in all native and reference isolates can be considered an extended set of core effectors for Xcc specifically as noted by Ryan et al. (2011).

While the function of all effectors is not known, those found in native strains are involved in host specificity, HR responses in certain crucifers and PTI evasion. Interestingly, a homologue of xopAG/avrGf1 was present in native strains but has only been characterised in X. citri to cause HR response in grapefruit (Gochez et al., 2015). Furthermore, hrpG and X transcriptional regulators of the T3SS were found in all Xcc. While the operon structure was conserved, sequence differences in native hrpX proteins (extra terminal amino acids) imply some adaptation to local host environments. Non-pathogenic Xanthomonas are routinely found to contain a reduced T3 effector repertoire, no pathogenicity island, but also contain the hrp transcriptional regulators (Merda et al., 2017; Lee, Yang & Huang, 2020).

A severely reduced effector repertoire was found in native Xmel strains along with hrpG and X genes. The T3SS pathogenicity island in Xmel was shown in previous work to have been lost at a later evolutionary event, despite the genus showing 3 ancestral acquisitions of this cluster (Merda et al., 2017). HGT events aided by mobile elements are mostly implicated for the gain and spread of these clusters, where they can explain relatively recent acquisition events and possibly account for in-planta co-inhabitation of non-pathogenic species (Lee, Yang & Huang, 2020). It has been proposed that Xcc obtained its T3SS clusters relatively recently with recombination with other species in the genus and that of certain Pantoea sp. strains were obtained from plasmids. Furthermore, the diversity of T3 systems in phytopathogenic bacteria have not been fully described and some without classified systems may simply have a different pathogenicity cluster type (Merda et al., 2017). Despite evidence for later acquisition in other members of the genus, native Xmel strains indicate that this still has not occurred and that no other T3SS type is present, adding further evidence to its non-pathogenic co-inhabitant status. A lack of T3SS and reduced effectors does not imply a lack of environmental fitness, as native Xmel strains possess other factors enabling effective in-planta colonization. Moreover, strains such as X. albilineans effectively colonizes xylem vessels despite its reduced T3SS and effectors (Merda et al., 2017).

Transfer and effective usage of pathogenicity islands have been demonstrated between pathogenic and non-pathogenic strains (Lee, Yang & Huang, 2020), highlighting the need to consider these commensal species as potential factors in the broad Xanthomonas gene pool. Additionally, multiple sources have shown that extracellular enzyme production can be regulated by the hrp regulon (Cesbron et al., 2015; Bansal, Kumar & Patil, 2020). While this supports the T2SS and excreted proteins as important pathogenicity factors, it also highlights the opportunistic lifestyle potential of Xmel and its ability to colonise actively infected/dead tissues. The entire xps and xcs T2SS gene clusters were found in all Xcc and Xmel genomes. Previous studies have found variable evidence of a link to AA substitutions in XpsD and T2SS substrate specificity in some Xanthomonas species (Lu et al., 2008), and in Xcc this was highly homologous but in Xmel there were many substitutions. Genes encoding PCWDEs (xylanases, pectinases, and glucanases) secreted by this system were found in both species, with Xcc strains containing more enzymes in each category as well as cellulases. Cellobiosidases, active in xylem targeting pathogens, were also found in both species. PCWDE differences are thought to reflect different metabolic niches during virulence lifestyles and to change with different host cell wall components (Ryan et al., 2011). For Xmel, these enzymes may be involved in nutrient acquisition in saprophytic/opportunistic lifestyles.

Comparative genomics revealing differences in T3SS and effector repertoire among other genetic differences are necessary but cannot replace functional studies. A virulence screen of native isolates revealed that Xmel strains did not produce symptoms and inoculated plants were similar to uninfected controls. Xcc strains were able to cause infection and some variations in symptomologies were observed. Thus, Xmel strains characterised in this study can be considered non-pathogenic within the context of the screening method and current genomic characterisation. Functional infection studies to elucidate its potential agricultural impact and its effect on co-infection, would be the subject of future research by our group. However, these types are severely underreported in relation to pathogenic strains and the current study demonstrates their genetic potential in agrochemical resistant populations. Additional focus to understand the ecological role of Xmel and other potential commensal species and their contribution to Xcc genetic diversity in infectious lifestyles is required.

Competitive capabilities of native Xanthomonas strains based on bacterial secretion system components

While collinearity of the X. citri pv. citri hylDB T1SS homologs (Fonseca et al., 2019) were maintained in native strains, no hemolysin-like toxin was found downstream. T1SS are primarily responsible for bacteriocins, adhesins, enzymes, and toxin secretion. This system is thought to play a role in the early stages of infection and might allow native isolates an advantage in competing phylloplane communities. NRPS BGC entF homologs (E. coli) predicted to produce enterobactin were more prevalent in Xmel but are common in Xanthomonas (Royer et al., 2013). Unidentified toxins inferred from non-homologous NRPS clusters may have phytotoxic and antimicrobial properties (Royer et al., 2013). Furthermore, the potential of some native strains as useful bioprotectant agents (Li et al., 2020) and the associated characterisation of novel NRPS clusters requires attention.

The variable T5SS in Xanthomonas mediates adhesion production. The yapH and xadAB genes are generally more prevalent in pathogenic Xcc (Bansal, Kumar & Patil, 2020) but were found in both native species, with Xmel additionally containing fhaBC genes. The yapH gene promotes attachment and reduced in-planta movement while xadA allows cells to establish within xylem structures (Alvarez-Martinez et al., 2021). The presence of the T4 pilus biogenesis cluster (T4P) is linked to plant colonization and possibly virulence as a preceding success factor (Fernandes et al., 2021). Most of this gene cluster was found in native isolates, but key genes such as pilI were not present in one Xcc, and pilX was only found in Xmel. While the potential for attachment to external and intracellular surfaces, and colonization potential is inferred from the T5SS and T4P, differences in gene profiles suggest disparities in these capabilities.

Interestingly, only Xmel contained the complete T6SS. This system delivers antagonistic effectors into competing bacteria (Bayer-Santos et al., 2018; Alvarez-Martinez et al., 2021). Contrastingly, previous research has observed a greater occurrence of T6SS in other pathogenic Xanthomonas (Bansal, Kumar & Patil, 2020; Alvarez-Martinez et al., 2021). Complex plant-microbe interactions, preferential survival in host tissues and other factors may compensate for native XCC lacking T6SS in competitive bacterial communities. The T4SS provides ecological advantages to bacteria by performing two major functions namely, conjugative transfer of DNA-protein complexes and, delivery of toxic proteinaceous effectors to competing bacteria. Except for DMCX, conjugative T4SS genes were identified in putative integrative mobile elements in all strains for both species A fully functional but variable T4SS relating to conjugative transfer is assumed in all native strains. Furthermore, homologs of the T4BSS Dot/Icm genes in some native Xcc strains have been associated with plasmids from X. vesicatoria (Jalan et al., 2011) and self-transmissible plasmids in other species (Voth, Broederdorf & Graham, 2012).

EPS and LPS biosynthetic gene clusters

Biofilm and its components promote Xanthomonas colonization of plant surfaces and tissues, as well as survival under adverse conditions and disease initiation (An et al., 2020). This extra polymeric substance is primarily composed of xanthan and is highly dynamic in response to signalling from bacterial cells induced by the environment. In Xcc, xanthan is linked to reduced Ca2+ related defence response in plants e.g., callose deposition (Aslam et al., 2008) and antagonizing plant triggered immunity by O-antigen detection of cell wall LPS (Girija et al., 2017). The gum operon, which regulates xanthan biosynthesis, was found in all Xcc and Xmel genomes, in addition to the xanthomonadin gene cluster. This corresponds to growth characteristics in the lab that are typical of this species (data not shown). LPS are important structural constituents of the bacterial outer membrane that influence resistance to antimicrobials and stresses. The biosynthesis of LPS is controlled by several gene clusters, all of which were conserved in the Xcc and Xmel genomes except for the wxc cluster. The latter is involved in O-antigen synthesis and varies between Xanthomonas species but shares a complex role in plant disease (Ryan et al., 2011; An et al., 2020). However, wxc gene mutants showed impaired T3SS function and disease development. The reduced wxc gene cluster in Xmel suggests an impaired ability to directly combat host defences in correlation to the lack of a T3SS. Interestingly, the wxcB gene in some native Xcc strains were not highly homologous to reference genes. This gene has been linked to O-antigen variability and structure which was also linked to reduced virulence in mutants (Park, Jung & Han, 2014). This in addition to the previously discussed variation in other virulence-related systems, points to disparities in virulent phenotypes which needs to be investigated in further functional studies.

Conclusion

This study represents the first comprehensive genome characterization of Xanthomonas campestris pv. campestris and co-inhabitant, Xanthomonas melonis. Both were isolated from crucifer leaves with black rot symptoms collected from copper agrochemical-impacted fields. While Xmel strains were obtained from the infected tissues, the results presented here reveal a lack of key genes known to be associated with pathogenicity and evasion of host defences. Regardless, its association with infections caused by Xcc is significant in the context of genomic reservoirs contributing to diversity within the genus and potential synergies between the two species. Characterization of hallmark features of the Xanthomonas genus such as secretion system variability, genome structural features, and the mobilome repertoire were presented in detail. A unique profile of T3SS effectors in native Xcc and Xmel suggests unique host adaptation traits which might influence host range or disease progression. This comprehensive genome study of native strains will contribute to a better understanding of the pathogenomics of Xcc and associated co-inhabitant species in Trinidad. The data presented here could be useful for future studies on key genomic features, selecting specific targets for virulence analysis and resistance breeding. The study also expands our understanding of combined genome characterization, virulence and resistance gene profiling for xanthomonads in the Caribbean region.

Supplemental Information

Supplemental Information 1 Supplemental Tables 1–11.

Click here for additional data file.

Supplemental Information 2 Assembled genomes of isolates used in this study.

Click here for additional data file.

We thank Professor Jeffrey Jones (University of Florida, Gainesville) for providing the Xcc BrA1 strain and Mr Omar Ali (University of the West Indies, St. Augustine) for critical reading of the manuscript.

Abbreviations

Xcc Xanthomonas campestris pv. campestris

Xmel Xanthomonas melonis

T1-6SS Type 1–Type 6 secretion system

PCWDE Plant cell-wall degrading enzymes

BGC Biosynthetic gene cluster

NRPS Non-ribosomal peptide synthetase

Additional Information and Declarations

Competing Interests

Author Contributions

Field Study Permissions

DNA Deposition

Data Availability

The authors declare that they have no completing interests.

Stephen D. B. Jr. Ramnarine conceived and designed the experiments, performed the experiments, analyzed the data, prepared figures and/or tables, authored, and approved the final draft.

Jayaraj Jayaraman conceived and designed the experiments, acquired research funds, authored or reviewed drafts of the paper, and approved the final draft.

Adesh Ramsubhag conceived and designed the experiments, acquired research funds, authored or reviewed drafts of the paper, and approved the final draft.

The following information was supplied relating to field study approvals (i.e., approving body and any reference numbers):

Diseased plant material from which Xanthomonas bacteria were isolated was sourced from farmers’ fields.

The following information was supplied regarding the deposition of DNA sequences:

The genomes of the native isolates from this study are available at GenBank for Ar1PC2, Ar1BCA1, Cf3C, Cf4B1, CaNP1C, CaNP1D, CaNP5B, CaNP6A, DMCX (PRJNA701249): JAFFQP000000000, JAFFQL000000000, JAFFQN000000000, JAFFQM000000000, JAFFQO000000000, JAFFQH000000000, JAFFQJ000000000, JAFFQI000000000, JAFFQK000000000.

The following information was supplied regarding data availability:

The raw data are available in the Supplemental Files.

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
