# Peer review of "Comparative genomics of the black rot pathogen Xanthomonas campestris pv. campestris and non-pathogenic co-inhabitant Xanthomonas melonis from Trinidad reveal unique pathogenicity determinants and secretion system profiles"

_PeerJ, doi:10.7717/peerj.12632_

## Round 0.1 · original submission · Major Revisions

The reviewers raised some concerns that must be addressed before the manuscript is considered for publication.

·

Basic reporting

The manuscript is well written. In fact, the introduction which focuses on the virulence and pathogenicity of members of the genus Xanthomonas is one of the most comprehensive I have ever read. It is succinct and well-formulated. The manuscript has a professional structure containing both tables and figures. The raw data are provided. Again the hypotheses are clearly presented.

Experimental design

The methodology described in this manuscript is based on bioinformatics, i.e. the authors use a number of computer programmes to analyse sequence data. This is a standard approach in this type of research and are described in sufficient detail to be repeated. The research question is well defined and relevant.

I have one question regarding Xmel. Very little information is provided on this species (in the introduction and discussion sections). The authors isolated it together with Xcc - is the association between Xmel and Xcc common, i.e. has it been reported elsewhere? The authors assume Xmel is non-pathogenic (and based on their results it lacks a number of key pathogenicity traits) but can one be sure without preforming inoculations? What would have been really interesting is to co-inoculate the species and determine if the severity is increased. This has been shown with other pathogens notably Ps. savastanoi pv. savastanoi.

Validity of the findings

The results are novel - the authors add to genome sequence information on additional strains of Xcc and of Xmel. The analyses of pathogenicity and virulence factors are comprehensive and provide novel results. The conclusions are well stated.

Additional comments

Line 108 - sentence is incomplete
Lines 477 and 716 - the word "species" is redundant - arboricola is the species
In the phylogenetic tree (Fig. 1) please indicate the type strains with the subscript T
Table 3 is confusing. Add the word "strains" in the first line, i.e. Xcc strains; Xmel strains
In Fig. 5A and 6A - what do the different colours mean?
Underneath Fig 6B indicate which strains are Xcc and Xmel
In Fig 8 what is A and B?
Again in Fig 9 indicate which strains are Xcc and which are Xmel

Reviewer 2 ·

Basic reporting

There is a scope and need to make the abstract and whole manuscript clearer and concise.

The title is also lengthier. There is also need a proper introduction to non-pathogenic Xanthomonas (NPX) in the introduction.

How authors are differentiating pesticides and fungicides? I can understand...copper, fungicide, bactericide...but what is the connection with fungicide...

In several instances, can finding results being repeated in discussion...

231 line...lastn?

Line 579 mentioning that native hrp ...were longer than references...is not clear...

The manuscript is written very roughly with lots of grammatical mistakes and incomplete sentences and typing errors (LN: 231, 430). Species name must be italic (LN: 57). Abbreviations are not introduced properly (LN: 67, 91-93).

Experimental design

1. How authors used bioanalyzer for quality check of genomic DNA...it is not clear...
2. Absence of a phylogenetic tree based on core-genome and authors have just relied on MLST data set.
3. Phylogenetic analyses (Figure 1) have missed many of the representative species of Xanthomonas genus. It needs to be revised according to the latest taxonomic classification of genus Xanthomonas.
4. LN: 223-224, Mention Blast coverage and percentage identity cut-off values used for the presence/absence of secretion system genes. Also, mention which type of Blast was used.
5. How meaningful is the findings related to genome dynamics (structure, IS elements, and plasmids) just based on draft genome sequences?

Validity of the findings

1 The authors have mentioned that the three X. melonis isolates to be non-pathogenic only based on the absence of T3SS but as the strains were isolated from infected plants there is a lack of pathogenicity studies and also HR response in the context of T3SS
2 The plasmid prediction profile done by the authors is very vague and does not add any relevance to the study. In some cases, (CaNP1C) coverage is very low. They might refer to more reliable plasmid prediction tools.
3 Data represented in Table 1 and LN: 278-279 should be verified.
4 LN: 170-171, which one of the sequenced strains was taken from the previous study? Mention in the manuscript.
5 LN: 183, what do mean by QC>30? Write clearly.
6 Revise Figure 5 and 6 legends to mention what the color codes represent.
7 Figure 11 – why the contig containing metB is missing in Xcc (B)? Is contig altogether missing?

Additional comments

None

---

## Round 0.2 · Minor Revisions

Please just address this last point raised by the reviewer.

·

Basic reporting

As this is a revision of a previous version, I reviewed the paper and then read the authors response to the reviewers comments. The only question I still have is that in the abstract the authors make the following statement: "Xmel strains were confirmed to be non-pathogenic by pathogenicity assays." - yet they do not include these trials in the materials and methods section. Unless this was done previously.

Experimental design

The experimental design answers the research questions posed and where necessary the information requested by the reviewers has been addressed.

Validity of the findings

The results are novel and supported by the data generated in their study.

Additional comments

In my opinion, the manuscript has addressed the concerned raised by previous reviewers and should now be accepted for publication.

regards
Teresa

---

## Round 0.3 · accepted · Accept

The authors have amended the manuscript following the reviewers' suggestions. Therefore, I recommend acceptance.